# GSP1-111 Modulates the Microglial M1/M2 Phenotype by Inhibition of Toll-like Receptor 2: A Potential Therapeutic Strategy for Depression

**DOI:** 10.3390/ijms251910594

**Published:** 2024-10-01

**Authors:** Ryeong-Eun Kim, Darine Froy Mabunga, Kyung-Jun Boo, Dong Hyun Kim, Seol-Heui Han, Chan Young Shin, Kyoung Ja Kwon

**Affiliations:** 1Department of Pharmacology, School of Medicine, Konkuk University, Seoul 05029, Republic of Korea; ryoung_2@naver.com (R.-E.K.); crushdfm@gmail.com (D.F.M.); bootony@naver.com (K.-J.B.); mose79@kku.ac.kr (D.H.K.); chanyshin@kku.ac.kr (C.Y.S.); 2Center for Neuroscience Research, Institute of Biomedical Science and Technology, Konkuk University, 120, Neungdong-ro, Gwangjin-gu, Seoul 05029, Republic of Korea; alzdoc@naver.com; 3Department of Neurology, Konkuk Hospital Medical Center, 120-1 Neungdong-ro, Gwangjin-Gu, Seoul 05030, Republic of Korea

**Keywords:** toll-like receptor 2, neuroinflammation, microglia, depression, therapeutics

## Abstract

Neuroinflammation plays a vital role in neurodegenerative diseases and neuropsychiatric disorders, and microglia and astrocytes chiefly modulate inflammatory responses in the central nervous system (CNS). Toll-like receptors (TLRs), which are expressed in neurons, astrocytes, and microglia in the CNS, are critical for innate immune responses; microglial TLRs can regulate the activity of these cells, inducing protective or harmful effects on the surrounding cells, including neurons. Therefore, regulating TLRs in microglia may be a potential therapeutic strategy for neurological disorders. We examined the protective effects of GSP1-111, a novel synthetic peptide for inhibiting TLR signaling, on neuroinflammation and depression-like behavior. GSP1-111 decreased TLR2 expression and remarkably reduced the mRNA expression of inflammatory M1-phenotype markers, including tumor necrosis factor (TNF)α, interleukin (IL)-1β, and IL-6, while elevating that of the M2 phenotype markers, Arg-1 and IL-10. In vivo, GSP1-111 administration significantly decreased the depression-like behavior induced by lipopolysaccharide (LPS) in a forced swim test and significantly reduced the brain levels of M1-specific inflammatory cytokines (TNFα, IL-1β, and IL-6). GSP1-111 prevented the LPS-induced microglial activation and TLR2 expression in the brain. Accordingly, GSP1-111 prevented inflammatory responses and induced microglial switching of the inflammatory M1 phenotype to the protective M2 phenotype. Thus, GSP1-111 could prevent depression-like behavior by inhibiting TLR2. Taken together, our results suggest that the TLR2 pathway is a promising therapeutic target for depression, and GSP1-111 could be a novel therapeutic candidate for various neurological disorders.

## 1. Introduction

Neuroinflammation is a well-established defense mechanism defined by the response of cells, such as neurons, astrocytes, and microglia, in the CNS [1]. Microglia and astrocytes are the main reactive macrophages in the CNS and play a critical role in the innate immune response. These cells respond as the first defender against exogenous toxic insults and proinflammatory stimuli such as cytokines, chemokines, and reactive oxygen species (ROS) [2,3,4]. Neuroinflammatory responses can either be beneficial or detrimental, depending on whether neuroinflammation, which is involved in the development of the normal brain, or a neuropathological process is triggered. These responses are considered defensive reactions, initially protecting the brain through the removal or inhibition of various pathogens, and are considered common pathophysiological mechanisms [5,6]. In other words, the inflammatory response is beneficial when it promotes tissue repair and removes cellular debris. However, an excessive and prolonged inflammatory response is detrimental and can lead to neurodegenerative diseases such as Alzheimer’s disease (AD), Parkinson’s disease (PD), and amyotrophic lateral sclerosis (ALS) [7]. Moreover, it has been reported that inflammatory responses are associated with depression [8]. Patients with major depression showed active inflammatory pathways, including increased levels of proinflammatory cytokines, chemokines, and adhesion molecules [9]. Similarly, higher levels of pro-inflammatory cytokines, associated with microglial activation, were found in animals with stress-induced depression [10].

Microglia are the principal innate immune cells in the brain and the primary responders against various pathologic insults, including pathogen-associated molecular patterns (PAMPs) and damage-associated molecular patterns (DAMPs) [7,11]. Microglia participate in housekeeping and host defense mechanisms by detecting environmental changes [12]. Neuroinflammation in the brain mainly constitutes microglial activation; they act as the first line of defense in various neurodegenerative diseases, such as AD and PD. Cellular receptors such as Toll-like receptors (TLRs) are expressed in the microglia, which produce proinflammatory cytokines, such as tumor necrosis factor (TNF) α and interleukin (IL)-1β, in response to various stimuli [12,13]. Microglia-mediated neuroinflammation is a common feature of various neurological disorders, including AD, PD, ALS, and multiple sclerosis (MS) [14,15]. In response to different stimuli, the microglia can be induced to form either the M1 pro-inflammatory phenotype or the M2 anti-inflammatory phenotype. M1 microglia release inflammatory cytokines and chemokines, which lead to neuronal inflammation and cell death. Conversely, M2 microglia set off anti-inflammatory mediators and initiate anti-inflammatory and neuroprotective effects, such as tissue repair and maintenance [16,17]. That is to say, neuroinflammation being mediated by microglia is basically a double-edged sword, mediating both harmful and beneficial effects on neurodegenerative diseases [18]. Therefore, the precise modulation of microglial activation through M1/M2 polarization is necessary for the normal functioning of microglia and maintaining brain homeostasis and could present a promising therapeutic strategy for neurological disorders.

TLRs belong to a family of pattern-recognition receptors that recognize PAMPs and DAMPs [13,19,20,21]. TLRs are single-pass type I transmembrane glycoproteins made up of leucine-rich repeat motifs at the N-terminus for ligand recognition, a single transmembrane helix, and a conserved cytoplasmic Toll/IL-1 receptor (TIR) domain at the C-terminus for intracellular signal transduction [22,23]. TLR families comprise the receptors TLR1 to TLR13, with 10 TLRs identified in humans and 12 TLRs in mice [21]. The plasma membrane mainly houses receptors like TLR1, TLR2, and TLR4–6, while TLR3 and TLR7–9 are mostly expressed within endosomes [24,25]. They play a vital role in the innate immune response, inflammation and immune cell regulation, and cell survival and proliferation [26,27,28,29]. Expressed in neurons, microglia, astrocytes, oligodendrocytes, and neural stem cells [30], TLRs can be activated by different DAMPs, including endogenous molecules released from damaged cells such as HMGB1, heat shock proteins, nucleic acid (dsRNA, dsDNA, and ssRNA), and mitochondrial DNA (mtDNA). In addition, TLRs are activated by the binding of PAMPs, which include lipopolysaccharide (LPS) from Gram-negative bacteria, flagellin, bacterial/viral nucleic acid including double-stranded RNA (dsRNA) and single-stranded RNA (ssRNA), CpG-rich DNA, β-glucan from fungus, and trehalose dimycolate (TDM) from *Mycobacterium* [31,32]. Once activated, these receptors initiate downstream signaling cascades such as the activation of myeloid differentiation factor 88 (MyD88), nuclear factor (NF)-κB, and mitogen-activated protein kinases (MAPKs). Among the TLRs, TLR2 and TLR4 are well known to be associated with disorders related to the brain, such as AD, PD, and stroke [33,34,35,36]. The expression levels of TLR2 and TLR4 were found to be upregulated in the brains of patients with AD and other neurodegenerative diseases such as PD. Moreover, the expression of TLR2 and TLR4 has been detected at increased levels in the microglial cells surrounding senile plaques in both the post-mortem brains of patients with AD and related mouse models [37,38].

Recently, the association between TLRs and depressive behavior has drawn considerable attention, given that TLR signaling is posited as a common inflammatory pathway. However, the inflammatory pathways mediating these depressogenic effects remain poorly elucidated. Depression is among the most widespread, incapacitating, and costly mental disorders worldwide. Reports show that the concentrations of inflammatory mediators, including IL-6, IL-1β, and TNFα, are elevated in major depressive disorder (MDD) [39,40].

GSP1-111 is provided by Genesen Co., Ltd. (Seoul, Korea) as a novel synthetic peptide. GSP1-111 is one of a peptide series designed by binding the cell-penetrating peptide (CPP) to eight sequences within the TIR domain of TIRAP, an adaptor protein of TLR. This peptide, consisting of 26 amino acids, penetrates into the cells and blocks the interaction between the intracellular Toll-IL-1 receptor (TIR) domain of TLR and the adaptor protein TIRAP, thereby inhibiting TLR-mediated downstream signaling and effectively disrupting TLR4 signal transduction [41,42,43]. GSP1-111 is a peptide drug candidate that can block excessive cytokine secretion by inhibiting the signaling pathway of the Toll-like receptor2/4 (TLR2/4).

In the present study, we demonstrated that GSP1-111, a novel TLR2 antagonistic peptide, could inhibit neuroinflammatory responses, revealing a potential role for TLRs in inflammatory depressive behavior. In addition, our findings may provide evidence supporting TLR2 as a potential therapeutic target in depression.

## 2. Results

### 2.1. GSP1-111 Decreases LPS-Induced Toxic Effects in BV2 Microglial Cells

To examine the toxic effects of GSP1-111 on BV2 microglial cells, an MTT assay was performed after 24 h. GSP1-111 treatment did not induce toxicity at the examined doses (Figure 1A). BV2 cells were treated with LPS (10 ng/mL) or PBS (vehicle) for 1 h following GSP1-111 or PBS (vehicle) treatment. After 24 h, 50 μg/mL MTT solution was added to treated cells to examine the protective effects mediated by GSP1-111. Our results revealed that 1 μM of GSP1-111 (96.86% ± 0.97, *p* < 0.0001 vs. LPS-treated group) increased cell viability (Figure 1A). In addition, GSP1-111 treatment significantly reduced the LPS-induced nitric oxide (NO) level (15.85 ± 0.086 μM, *p* < 0.0001 vs. vehicle control) in a concentration-dependent manner (14.11 ± 0.31, 8.524 ± 0.11, and 4.32 ± 0.17 μM, *p* < 0.0001 vs. LPS-treated group) in BV2 microglial cells (Figure 1B).

### 2.2. GSP1-111 Reduces mRNA and the Protein Expression of TLR2 in BV2 Microglial Cells

We next determined whether GSP1-111 could inhibit LPS-stimulated TLR 2 and TLR4 activation. Accordingly, the cells were treated with TLR2 antagonist (C29, 50 μM) and TLR4 antagonist (TAK-242, 0.5 μM) and subsequently compared with GSP1-111-treated cells. After 24 h, the released NO was measured and compared with the GSP1-111-treated group. The LPS-induced increase in NO (10.02 ± 0.29 μM) was decreased by GSP1-111 treatment (2.05 ± 0.25 μM), similar to TLR4 (TAK-242) and TLR2 (C29) antagonist-treated groups (2.02 ± 0.16 μM, 4.62 ± 0.90 μM, *p* < 0.0001 vs. LPS-treated group, respectively) (Figure 2A).

To assess whether GSP1-111 has inhibitory effects on LPS-induced TLR2 and TLR4 expression, we examined mRNA expression levels of TLR2 and TLR4 in BV2 microglial cells. BV-2 microglial cells were treated with 0.01, 0.1, and 1 µM of GSP1-111 and 10 ng/mL of LPS. After 24 h, RNA was isolated from cultured BV-2 cells using TRIzol reagent. The isolated RNA was quantified, and reverse transcription-quantitative PCR (RT-qPCR) was performed to synthesize cDNA (0.5 μg). Using TLR2 and TLR4 primer sets, PCR was employed to measure the expression levels of TLR2 and TLR4 mRNA. Based on our findings, LPS treatment increased TLR2 (4.09-fold, *p* < 0.0001 vs. vehicle control) and TLR4 (1.22-fold, *p* < 0.0001 vs. vehicle control) mRNA expression; treatment with GSP1-111 decreased TLR2 mRNA expression (1.16-fold, *p* < 0.0001 vs. LPS-treated group) but not TLR4 mRNA expression (Figure 2B). Similar to the results of mRNA expression, LPS treatment increased the protein expression of TLR2 (2.52-fold, *p* < 0.05 vs. vehicle control), while GSP1-111 decreased the TLR2 protein expression that was increased by LPS treatment (1.17-fold, *p* < 0.05 vs. LPS-treated group, respectively) (Figure 2C). These results suggested that GSP1-111 has an inhibitory effect on TLR2.

### 2.3. GSP1-111 Treatment Inhibits LPS-Induced Neuroinflammation in BV2 Microglial Cells

We found that GSP1-111 treatment reduced TLR2 expression, a common inflammatory molecule in BV-2 microglial cells. Accordingly, we examined the protective effects of GSP1-111 against LPS-induced neuroinflammation in BV2 microglial cells. GSP1-111 treatment effectively lowered the expression of various inflammatory cytokines, including IL-1β, TNFα, and IL-6 (M1-specific markers), compared with the LPS-induced increase in levels, but GSP1-111 alone did not affect the expression of the cytokines we identified. Additionally, GSP1-111 treatment could restore the LPS-induced decreased expression of IL-10 and arginase-1 (M2-specific markers) (Figure 3A). We also investigated the effects of GSP1-111 on the protein expression of M1 markers (iNOS and COX-2) and M2 markers (CD206 and IL-10). As a result, GSP1-111 treatment significantly decreased the protein expression of M1 markers (1.05-fold, 1.03-fold, *p* < 0.01, and *p* < 0.001 vs. LPS-treated group, respectively) increased by LPS (4.34-fold, 3.24-fold, *p* < 0.01, and *p* < 0.001 vs. vehicle control, respectively), and increased the protein expression of the M2 marker, CD206 (1.36-fold, *p* < 0.001 vs. LPS-treated group) decreased by LPS (0.14-fold, *p* < 0.01 vs. vehicle control) (Figure 3B). In addition, even when GSP1-111 and LPS were treated together, the same effects as with the pre-treatment were obtained. These findings suggest that GSP1-111 inhibited LPS-induced neuroinflammation by altering microglial polarization.

### 2.4. GSP1-111 Decreases LPS-Induced MAPKs and Akt Phosphorylation

LPS induces the activation of various signaling molecules. Aiming to examine the signaling pathways traversed by the processes of LPS-induced neuroinflammation, we observed changes in the expression and phosphorylation of various MAPKs (ERK, p38, and JNK) and Akt using Western blotting. LPS treatment significantly enhanced p-ERK, p-JNK, p-p38, and p-Akt phosphorylation by approximately 2.5~3-fold, whereas GSP1-111 treatment reduced the phosphorylation of MAPKs and Akt to the levels observed in the vehicle control group (Figure 4).

### 2.5. GPS1-111 Decreases the Immobility Time in LPS-Induced Depressive Mouse Model

Accumulated evidence supports the possible role of neuroinflammation in depressive disorder [41,42]. The levels of proinflammatory mediators are reportedly increased in the blood and cerebrospinal fluid of patients with depression. Longitudinal meta-analyses and experimental studies have demonstrated that the inflammatory response may induce the beginning, upkeep, or relapse of depressive features in MDD. In addition, preclinical and clinical studies have shown that the dysregulation of TLR expression and signaling can be associated with depression.

To determine the effects of GSP1-111 on inflammation-related neurological disorders, we used an LPS-induced depression mouse model, wherein the mice were subsequently subjected to a forced swimming test (FST) and a tail suspension test (TST) according to an experimental timeline (Figure 5A). After administering GSP1-111 and PBS (vehicle) subcutaneously for 5 days, LPS and saline (control) were given intraperitoneally. An FST and a TST were performed to examine the antidepressant effects of GSP1-111 24 h after LPS injection. During the FST, LPS-administered mice (148.0 ± 8.24, *p* < 0.0001) presented an increase in immobility time when compared with those administered the vehicle (saline, 68.38 ± 10.86 s). Conversely, the GSP1-111-treated group (104.78 ± 8.24, *p* < 0.01) showed a significant decrease in the immobility time when compared with the LPS-administered mice (Figure 5B). Similar to the FST results, during the TST, it was also shown that the immobility time was increased in the LPS-administered mice (172.63 ± 17.18 s, *p* < 0.01) compared to the vehicle control group (saline), while GSP1-111-treated mice (131.13 ± 19.69 s, *p* < 0.05) showed a reduced the immobility time compared to the LPS-administered mice (Figure 5C). During the experimental period, no difference in body weight was noted between the experimental groups (Figure 5D). Accordingly, these results revealed that GSP1-111 significantly reduced LPS-induced depression-like behavior without incurring toxicity.

### 2.6. GSP1-111 Significantly Prevents LPS-Induced M1 Polarization in the Brain

To identify the effects of GSP1-111 in the brains of LPS-administered mice, we euthanized the mice and harvested their brains. We examined the gene expression profiles of different inflammatory cytokines, such as M1 markers, including IL-1β, TNFα, IL-6, inducible nitric oxide synthase (iNOS), and cyclooxygenase-2 (COX-2), along with the gene expression levels of IL-10 and arginase-1 as M2 markers using RT-qPCR. We found that LPS administration increased the transcription of M1-specific inflammatory genes while decreasing that of the M2-specific genes, IL-10 and arginase-1, in the frontal cortex of the brain (Figure 6).

GSP1-111 treatment prevented an inflammatory response in BV2 microglial cells (Figure 3). Therefore, we investigated whether GSP1-111 treatment can suppress the LPS-induced M1/M2 phase changes in the mouse brain. GSP1-111 treatment significantly decreased the expression of various M1 markers, including IL-1β, TNFα, and IL-6 (26.7, 47.2, and 51.53%, *p* < 0.05, vs. LPS-injected group, respectively), which were increased by LPS administration, to levels observed in the vehicle control. Additionally, the mRNA expression of IL-10 and arginase-1 (75.7 and 57.5%, *p* < 0.0001 vs. LPS-injected group, respectively), which were decreased by LPS administration (79.8% and 82.5%, *p* < 0.01 vs. vehicle control, respectively), were restored following GSP1-111 treatment (Figure 6). These results indicate that GSP1-111 inhibited LPS-induced neuroinflammation by altering microglial polarization.

### 2.7. GSP1-111 Decreases Microglial Activation in LPS-Injected Mouse Brain

To examine the effects of GSP1-111 on LPS-induced microglial activation, we performed immunohistochemistry tests using perfused brain tissue. Following the FST, the experimental mice were euthanized, then the brains were perfused and sectioned for immunohistochemical staining. Each brain section was incubated with the Iba-1 antibody (microglia-specific marker). The brain tissue of LPS-injected mice exhibited increased Iba-1 positive cells (relative fluorescent intensity, 8.75 ± 0.82, *p* < 0.0001 vs. vehicle control), whereas GSP1-111 treatment suppressed this increase in Iba-1 staining (2.33 ± 1.22, *p* < 0.0001 vs. LPS-treated group), thereby preventing the associated morphological changes (Figure 7). Collectively, these results suggest that LPS can induce depressive behavior and inflammation in microglial cells and that GSP1-111 treatment could inhibit neuroinflammation-mediated depressive behavior.

### 2.8. GSP1-111 Inhibits Brain Expression of TLR2 Protein

We measured TLR2 expression to determine whether GSP1-111 prevents increased TLR2 expression and neuroinflammation after LPS injection. After performing behavioral tests, the mice were euthanized and perfused. The perfused mouse brains were dissected and used for an mRNA and protein expression analysis of TLR2 and TLR4. LPS administration increased the mRNA and protein expression of TLR2 in the frontal cortex of the brain (9.78-fold and 1.22-fold, *p* < 0.0001 and *p* < 0.01 vs. vehicle control, respectively), whereas GSP1-111 significantly decreased the TLR2 mRNA and protein expression levels (6.2-fold, 0.86-fold, *p* < 0.01 and *p* < 0.05 vs. LPS-injected group, respectively) (Figure 8A,B). GSP1-111 did not decrease the mRNA and protein levels of TLR4 expression. Additionally, immunohistochemistry data showed that GSP1-111 reduced TLR2 immunoreactivity in CD11b-stained microglia (Figure 8C). These results indicated that GSP1-111 could prevent the inflammatory response via TLR2 expression in an LPS-induced depression model.

## 3. Discussion

The major finding of the present study is that the synthetic peptide GSP1-111 blocks neuroinflammatory responses, thereby suppressing the increase in M1-specific markers by decreasing TLR2 expression; this TLR2 inhibition potentially alleviates LPS-induced depressive behavior. Neuroinflammatory responses may be either advantageous or damaging, depending on the mechanisms associated with neuroinflammation, which is known to be involved in both normal brain development and neuropathological processes. Prolonged inflammatory responses are critical pathological features of various neurological disorders, including traumatic brain injury, MS, epilepsy, psychological disorders, and neurodegenerative diseases. Therefore, neuroinflammation is a common mechanism associated with ischemic, degenerative, traumatic, demyelinating, epileptic, and psychiatric pathologies [3,30,43,44]. Notably, all CNS cell types can contribute to the inflammatory process. Microglia are innate immune cells of the CNS, acting as first-line defenders in the brain by eliminating dead cells, repeating synapses, protein aggregates, and harmful pathogens, including other particulate and soluble antigens [17,45]. Under physiological conditions, the microglia play a critical role in maintaining cellular homeostasis and facilitating the proliferation of neural precursor cells during brain development [46,47]. After CNS injury, the microglia activate and secrete proinflammatory mediators, which enhance the cytotoxicity of the healthy neural tissue around them; in turn, inflammatory factors released by dead or dying neurons amplify the prolonged activation of microglia, causing progressive neuronal loss. These microglial changes have been detected in various neurodegenerative diseases, such as AD, PD, Huntington’s disease (HD), MS, and ALS [17,46]. Therefore, resolving microglia-mediated neuroinflammation during disease pathology may represent a novel treatment strategy to reduce brain degeneration. Our results have demonstrated that GSP1-111 is a novel candidate for treating neurological disorders via microglia-mediated neuroinflammation.

GSP1-111 was supplied by Genesen Co., Ltd. (Seoul, Korea) as a novel synthetic peptide. GSP1-111 is one of a peptide series consisting of 26 amino acids, designed to penetrate cells by being tagged with a cell-penetrating peptide (CPP). CPPs are a specialized group of peptides capable of crossing cell membrane bilayers without causing significant lethal membrane damage. Blood-brain barrier (BBB) penetration is a pivotal challenge in the development of treatments for neurological disorders, particularly nucleic acid-based medicines. CPPs stand out as the most promising technology for enabling macromolecules, such as peptides, to penetrate the BBB. Some studies have shown that CPPs can reach the brain parenchyma, both in vivo and in vitro [48,49]. While we do not yet have direct evidence that GSP1-111 can cross the BBB, its clear effect on neuroinflammation suggests the efficient permeation of GSP-111 through the BBB. Further studies on BBB penetration, including brain pharmacokinetics, are needed.

TLRs play various important roles in mediating the innate immune response. TLRs are widely distributed in organisms, especially in the CNS, and are expressed in neurons, astrocytes, and microglia. In particular, in the microglia and astrocytes, TLR expression is induced by various stimuli, including inflammatory stimuli such as LPS. McCarthy GM et al. demonstrated that in response to LPS, microglial cells highly expressed TLR2 and TLR4, and TLR3 expression was found to be highest in astrocytes, also increasing in response to LPS. These findings are consistent with several studies showing that TLR3 is highly expressed in astrocytes [9,50,51]. TLRs, especially TLR2 and TLR4, were found to participate in neurodegenerative diseases such as AD, PD, and ALS, as well as in neuropsychiatric diseases such as depression [52,53]. In microglia, TLR activation can be induced by different stimuli, including hypoxia, LPS, poly[I:C], amyloid beta, and α-synuclein [54,55,56,57,58]. TLR2, as well as TLR4, are induced by Gram-negative endotoxins such as LPS [59,60]. TLR2/4 is expressed in the microglial cells, and repeated stress results in increased microglial activation through TLR2/4 expression and causes depressive behavior [61]. In addition, Murayama S et al. identified the high expression of TLR2 in the microglia, where activation of the microglia causes severe fever and inflammatory responses [62]. TLR activation in the microglia might regulate the activity of these cells, which leads to either beneficial or detrimental effects on the surrounding cells, including neurons and astrocytes. Therefore, TLRs may be a molecular link that modulates neuroinflammation and neurodegenerative diseases. In the present study, we aimed to identify the modulators of TLRs in the brain. Our findings revealed that GSP1-111 reduces LPS-induced TLR2 expression; thus, GSP1-111 can regulate TLRs, mainly through the TLR2 found in the brain.

Owing to the dual roles mediated by microglia in neuroinflammation through the microglial phenotype, the microglia can diverge into either the M1 pro-inflammatory phenotype or the M2 anti-inflammatory phenotype, depending on microenvironmental disturbances [63]. Inflammatory mediators, such as IL-1β, TNFα, and iNOS, are released by the M1 microglia, stimulating inflammation and neurotoxicity, while anti-inflammatory mediators, such as IL-10 and arginase-1, are released by the M2 microglia, thereby inducing anti-inflammatory and neuroprotective effects. Accordingly, it is evident that the microglia are involved in the pathogenesis of neurodegenerative disorders, thereby suggesting that microglia may be seen as a double-edged sword in neurodegenerative diseases [18]. Modulating microglial polarization from M1 to M2 has promising therapeutic prospects in treating neurodegenerative diseases [14]. Cui W et al. have reported that the inhibition of TLR4 induces M2 microglial polarization and provides neuroprotective effects in AD, and that a TLR4-specific inhibitor, TAK-242, could significantly improve neurological function in a mouse AD model [64]. Herein, our results demonstrated that LPS significantly increased the shift of microglia from the M1 to the M2 phenotype, and GSP1-111 could regulate microglial M1/M2 polarization. Therefore, our data suggest that GSP1-111 acts as a potential modulator of microglial M1/M2 polarization and is a promising candidate for treating various neuroinflammation-related neurological disorders.

TLR activation induces TNFα and IL-1β release and is a key receptor in proinflammatory signaling. Microglia express a variety of TLRs, and, once activated, trigger diverse signal transduction pathways, including MAPK and phosphoinositide-3 kinase/protein kinase B (PI3K/AKT), thereby activating NF-κB. Consequently, NF-κB activation induces the production of proinflammatory cytokines, chemokines, iNOS, and COX-2, resulting in neuroinflammation. Moreover, LPS stimulation leads to the serine phosphorylation of Akt, mediated via TLR4. Our data showed that LPS increased the phosphorylation of MAPKs (ERK1/2, p38, and JNK) and Akt, whereas GSP1-111 decreased their phosphorylation levels.

Depression is among the most prevalent and disabling mental disorders, inflicting a substantial economic burden [39]. Several studies have linked the activation of the inflammatory system to depression. Elevated concentrations of inflammatory mediators have been noted in the blood and cerebrospinal fluid of patients with depression [40,65]. The inflammation process affects the initiation, severity, and symptoms of MDD through different interconnecting pathways involved in the modulation of neurogenesis, dopaminergic and serotonergic metabolism, and hypothalamic-pituitary-adrenal (HPA) axis activation. In the case of MDD, depressogenic effects influence the cellular immune response by activating the sympathetic nervous system and HPA axis, thereby boosting central and peripheral inflammatory mediators. Several experimental studies and systemic reviews suggest that levels of IL-6, IL-1β, C-reactive protein (CRP), TNFα, and the IL-1 receptor antagonist (IL-1Ra) are elevated during MDD [42,66]. However, the mechanism underlying these depressive phenotypes remains poorly clarified, and TLR signaling has been suggested as a potential inflammatory pathway. In the present study, our results demonstrated that LPS induced depression-like behavior through TLR2 expression related to microglia with the M1 phenotype, whereas GSP1-111 inhibited TLR2 expression and increased the number of microglia exhibiting the M2 phenotype. These results suggest a potential role for TLR2 in microglial M1/M2 polarization in LPS-induced depressive behavior. Moreover, TLR2 is associated with depression and conditions such as MDD, and TLR2 inhibition may be a viable target for MDD treatment.

Depression is a common condition that significantly impairs psychosocial functioning and lowers the quality of life [67,68]. Women are more vulnerable than men to stress-related psychopathologies such as depression, with twice the incidence rate and four times the risk of experiencing recurrent depressive episodes. Among the various symptoms associated with mood disorders in women, depression is reported to be the most severe. Chronic stress is known to be a major risk factor for major depressive disorder (MDD), affecting the immune system and activating the microglia in the mPFC. Our experimental model is an acute depression model based on inflammation. Initially, we tested the GSP1-111 compound on male animals to evaluate its efficacy against depression. It is plausible that the differences between males and females may not be evident in the inflammatory-induced depression animal model that we used. There are reports indicating behavioral differences and varying drug responses between males and females in long-term depression models and human depression studies [69,70,71]. Recent studies have shown that both male and female mice exposed to chronic stress exhibit depression-like behaviors, but only female mice display persistent depression-like behaviors [72]. Moreover, this persistent depressive behavior in female mice has been linked to TLR4 and microglial activation. This study supports the hypothesis that TLR4 in the microglia may regulate the sex differences in persistent depression-like behaviors in females. Additionally, TLR4 knockout mice exhibited pronounced depression-like behavior, while TLR2 knockout mice showed significant impairment in recovery from depression in the male mice [9]. There is also evidence that anxiety and social avoidance are induced by microglial activation through TLR2/4 in a repeated social defeat stress-induced depression model [61]. Therefore, future studies are needed to investigate inflammation and sex-related differences in neurological disorders such as depression.

## 4. Materials and Methods

### 4.1. Materials

The materials used in the present study were as follows: Dulbecco’s modified Eagle’s medium (DMEM), penicillin-streptomycin (P/S), 0.25% trypsin-ethylenediaminetetraacetic acid (EDTA), and fetal bovine serum (FBS) were sourced from Gibco BRL (Grand Island, NY, USA). Dimethyl sulfoxide (DMSO) was procured from Invitrogen (Carlsbad, CA, USA), and the ECL Western blotting detection reagent was purchased from iNtRON Biotech (Seoul, Republic of Korea). LPS (O111:B4) and anti-β-actin were purchased from Sigma-Aldrich (St. Louis, MO, USA), while C29 (TLR 2 antagonist) and TAK-242 (TLR4 antagonist) were purchased from MedChemExpress (MCE, Monmouth Junction, NJ, USA). Anti-phospho-ERK, anti-phospho-JNK, an-ti-phospho-p38, anti-phospho-Akt, anti-ERK, anti-JNK, anti-p38, and anti-Akt antibodies were purchased from Cell Signaling Technology (Danvers, MA, USA), the anti-TLR2 antibody was purchased from Invitrogen (Carlsbad, CA, USA), and anti-TLR4 antibodies were purchased from Novus Biologicals (Littleton, CO, USA). GSP1-111 is a novel synthetic peptide obtained from Genesen Co., Ltd. (Seoul, Republic of Korea).

### 4.2. Animals

All procedures involving animals in this study were approved by the animal care and use guidelines approved by the Animal Care and Use Committee (IACUC) of Konkuk University (Permit Number: KU21205). All experiments, including treatment, anesthesia, and euthanasia, were conducted in accordance with ARRIVE guidelines to minimize the number of animals used and any pain or stress they might experience. A total of 19 male ICR mice (20–30 g; 4 weeks old) were purchased from Orient Bio (Gyeonggi, Korea) and used for the behavior experiments. Male ICR mice (20–30 g; 4 weeks old) were purchased from Orient Bio (Gyeonggi, Republic of Korea) and used for the experiments. All mice were housed in a room maintained at an ambient temperature of 23 ± 0.5 °C, with a relative humidity of 55 ± 2%, on an automatically controlled 12 h:12 h light/dark cycle (lights on at 7:00). All mice had free access to food and water. Based on previous in vivo experiments, a sample size calculation (power = 1; α = 0.05) estimated 9 animals per group using G power version 3.1.9.7. For IHC staining, 5 animals were randomly assigned to each group and 5 animals/group were used for inflammatory gene and protein expression measurement (effect size = 3.47, power = 0.99). The animals and samples were coded, and all researchers were blinded to the treatment groups until the end of the data analysis.

### 4.3. Cell Line and LPS Treatment

We employed BV-2 murine microglial cells (we received these cells from Professor Eun Hye Joe, Ajou University, in South Korea), which were maintained in high-glucose DMEM, supplemented with 5% heat-inactivated FBS, 100 units/mL penicillin, 100 mg/mL streptomycin, and 2 mM glutamine (GibcoTM, Grand Island, NY, USA). Cells were seeded at 2 × 10^5^ cells/mL in 24- or 6-well plates (TPP, Trasadingen, Switzerland) and incubated at 37 °C in a humidified 5% CO_2_ incubator. Cultured cells were treated with 0.01–1 μM GSP1-111 or the vehicle for 1 h and then treated with 10 ng/mL of LPS for the indicated time, with or without GPS1-111.

### 4.4. Determination of Nitrite Concentration

NO production was determined by measuring nitrite production, as previously described [73]. Briefly, Griess reagent was freshly prepared by mixing equal volumes of 0.1% naphthylethylenediamine dihydrochloride and 1% sulfanilamide in 5% phosphoric acid, then this was added to the spent culture media for 5 min. The absorbance of these mixtures was measured at 540 nm using a UV spectrophotometer (Spectramax 190 molecular device, Palo Alto, CA, USA), and NO was quantified using a sodium nitrite (1–40 µM) standard curve.

### 4.5. Reverse Transcription-Quantitative Polymerase Chain Reaction (RT-qPCR)

The expression levels of *Il-1β*, *Il-6*, *TNFα*, *TGF-β*, *iNOS*, *COX-2*, *Tlr2*, and *Tlr4* in treated BV-2 microglial cells were determined by RT-qPCR. RNA was extracted using the TRIzol reagent (Invitrogen, Carlsbad, CA, USA), and the RNA concentration was measured using a spectrophotometer (Nanodrop Technologies, Wilmington, NC, USA). cDNA synthesis was performed using 0.5 µg of total RNA and an RT reaction mixture containing RevertAid Reverse transcriptase, reaction buffer (Thermo Fisher Scientific, Waltham, MA, USA), and dNTP (Promega, Madison, WI, USA). The gene-specific primer sequences used in this study are listed in Table 1.

RT-qPCR was performed using the BlasTaqTM 2x qPCR Master Mix (Applied Biological Materials Inc., Richmond, BC, Canada) on an ABI 7500 Real-Time PCR system (Applied Biosystems, Waltham, MA, USA). All assays, including those for controls, were performed in triplicate. The expression levels of 18s rRNA were used as internal controls, and the relative expression of each transcript was calculated using the 2^−ΔΔCT^ formula to determine the fold change, as described in the ABI user guide.

### 4.6. Western Blot Analysis

After treatment with GSP1-111 and LPS, cells were harvested in radioimmunoprecipitation assay buffer consisting of 2 mM EDTA, 0.1% (*w*/*v*) sodium dodecyl sulfate (SDS), 50 mM Tris-HCl, 150 mM sodium chloride, 1% Triton X-100, and 1% (*w*/*v*) sodium deoxycholate. Total protein was quantified using a bicinchoninic acid (BCA) assay kit (Thermo Fisher Scientific, Waltham, MA, USA) and boiled for 5 min at 100 °C. In total, 10 µg of protein from each sample was loaded onto a 10% SDS-polyacrylamide gel and subjected to electrophoresis (SDS-PAGE) at 100 V for 120 min. The separated proteins were then transferred to nitrocellulose membranes for 90 min, and the blots were blocked with 5% skim milk in Tris-buffered saline with 0.1% Tween 20 (TBST) for 1 h at 25 °C. Subsequently, the blots were washed with TBS-T and incubated with the appropriate primary antibodies, namely, β-actin (1:40,000), p-ERK/ERK/p-p38/p38/p-JNK/JNK/p-Akt/Akt (1:1000), CD206/IL-10/iNOS/COX-2 (1:1000) and TLR2/TLR4 (1:1000), overnight at 4 °C, and then washed three times and incubated with horseradish peroxidase (HRP)-conjugated secondary antibodies (Life Technologies, Carlsbad, CA, USA) at room temperature for 60 min. The bands were detected using an enhanced chemiluminescence detection system (iNtRON Biotech, Seoul, Korea) and visualized using a LAS-3000 image detection system (Fuji, Japan). The bands were quantified using the ImageJ system (National Institutes of Health, Bethesda, MD, USA) and β-actin was used as the loading control.

### 4.7. Measurement of Cell Viability

BV-2 microglial cells were cultured in 24-well plates, with each treatment group represented in 4 wells. After treatment with GSP1-111 (0.01 0.1, and 1 µM), the cells were further incubated with 10 ng/mL LPS for 24 h and then evaluated for cell viability using the 3-[4,5-dimethylthiazol-2-yl]-2,5-diphenyl-tetrazolium bromide (MTT) assay. MTT is a water-soluble tetrazolium salt reduced by metabolically viable cells to a colored, water-insoluble formazan salt. MTT (5 mg/mL) was added to the cell culture medium, then the cells were incubated at 37 °C for 2 h in a 5% CO_2_ atmosphere. The MTT-containing medium was then replaced with DMSO, and the absorbance was measured at 570 nm using a microplate reader (Spectramax 190, Molecular Devices, Palo Alto, CA, USA). The percentage of surviving cells was calculated and compared with that of the untreated control group (untreated cells).

### 4.8. LPS-Induced Depression Model

ICR mice (10-week-old) were purchased from Orient Bio (Seoul, Republic of Korea) and were habituated for five days in a designated animal room with automated systems to a light-on and light-off cycle (lights on, 24:00; lights off, 12:00) and maintained at a constant temperature (23 ± 2 °C) and humidity (50 ± 10%). The mice were provided with food and water ad libitum. All procedures, including those for animal treatment and maintenance, were performed following the principles of Laboratory Animal Care (NIH publication NO.85-23, revised 1985) and implemented according to the guidelines of the Animal Care and Use Committee of Konkuk University, Republic of Korea (KU20195).

After 7 days of habituation, the mice were injected with 10 mg/kg of GSP1-111 or PBS for 5 days subcutaneously. On day 5, the mice were injected with 0.8 mg/kg of LPS or saline intraperitoneally at a volume of 10 mL/kg [74,75]. This dose of LPS resulted in an increased immobility time 24 h after the LPS challenge, without reducing locomotor activity [76,77,78]. The behavioral test was performed 24 h after LPS injection.

### 4.9. Forced Swimming Test

To examine the effect of GSP1-111 on depressive behavior, a forced swim test (FST) was performed and 10 animals were used per experimental group [9,79]. A transparent plexiglass cylinder (15 cm diameter; 25 cm height) was filled with water (25 ± 1 °C; 15 cm depth). Each mouse was placed on the water surface and allowed to swim or float for 6 min while a video camera recorded the whole trial. Then, the subject mice were carefully removed from the cylinder and dried with a towel before placing them back in their home cages. The water was changed after each trial. The behavior test was conducted between 12:00 and 14:00 h. The first 2 min of a total time of 6 min was used for habituation. The immobility time during the last 4 min of the trial was measured by a highly experienced observer who was blinded to the experimental conditions. Immobility was defined as maintaining submergence in the water with only minimal movements to keep their heads above the water.

### 4.10. Tail Suspension Test

To examine the antidepressant activity of GSP1-111, a tail suspension test was performed for 6 min as described previously [9]. Ten animals per experimental group were used. Both acoustically and visually isolated mice were individually suspended 30 cm above the ground by adhesive tape placed over 1 cm from the tip of the tail. Mice were suspended for 6 min, and the immobility time was measured using a digital stopwatch during the last 4 min by an experienced observer (blinded to the experiment). When the subject mice hung passively and were completely motionless, they were considered immobile. Immobility was defined as the absence of limb movement.

### 4.11. Immunohistochemistry

To determine the effect of GSP1-111 on microglial activation, ICR mice were administered GSP1-111 (10 mg/kg) or a vehicle (PBS) daily for 5 days. On the last day, the experimental animals were intraperitoneally administered LPS (0.8 mg/kg) or saline. After 24 h, the mice were subjected to transcardial perfusion, and their brains were fixed using a 4% paraformaldehyde (PFA) solution. The perfusion-fixed brain tissues were frozen and sectioned using a cryostat (30-μm thickness). Each brain section was processed for immunohistochemical staining. The brain sections were rinsed with PBS, permeabilized with PBS containing 0.3% TritonX-100 and 0.5% BSA for 1 h at 25 °C, and then blocked using blocking buffer (1% BSA, 5% HS in PBS) for 1 h at 25 °C. The tissue sections were incubated with the appropriate primary antibody against Iba-1 (microglia, 1:500, Wako Chemicals, Osaka, Japan) and CD11b (activated microglia, 1:100, Millipore, CA, USA), TLR2 (1:500, Invitrogen, Carlsbad, MA, USA), and CD206 (M2-specific marker, 1:100, Abcam, Cambridge, UK) at 4 °C overnight. The next day, the tissue sections were rinsed with washing buffer containing 1.5% HS and 0.1% Triton X-100 in PBS and washed three times for 10 min each. The tissue sections were then incubated at room temperature with secondary antibodies conjugated with Alexa488 in a blocking buffer for 2 h at 25 °C. After three washes with PBS, the tissue sections were incubated with DAPI to allow nuclear staining, mounted using Vectashield (Vector Laboratories, Burlingame, CA, USA), and then observed using a confocal microscope (LSM900, Carl Zeiss, Aalen, Germany). Images were captured in three random areas per section, with five brains sampled per group.

### 4.12. Statistical Analysis

Data were analyzed using GraphPad Prism version 8.4.3 software (GraphPad Software, Inc., San Diego, CA, USA). All data were expressed as the mean ± standard deviation (SD). Statistical comparisons were performed using a one-way analysis of variance (ANOVA) test, followed by Dunnett’s multiple comparison test as a post hoc test, and a value of *p* < 0.05 was considered significant.

## 5. Conclusions

Taken together, our findings demonstrate that GSP1-111, a novel synthetic peptide for suppressing TLR2 signaling, inhibits neuroinflammatory responses through M2 microglial polarization and decreases LPS-induced depression-like behavior. Accordingly, we suggest that TLR2 could be a potential therapeutic target in depression treatment. Moreover, GSP1-111 is a potential treatment option for inflammatory neurological disorders. However, further studies are warranted to elucidate the role of TLR2 in these diseases and to confirm the pathophysiological role of TLR2 in neuropsychiatric disorders.

## Figures and Tables

**Figure 1 ijms-25-10594-f001:**
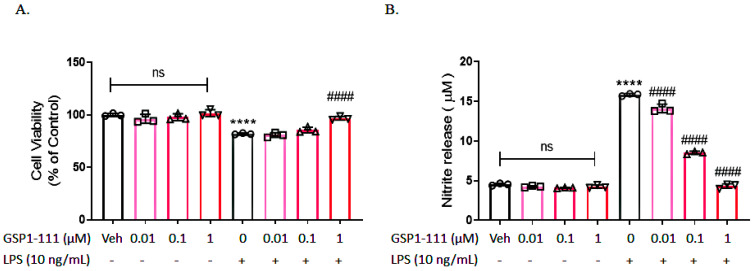
GSP1-111 does not induce toxic effects in BV2 microglial cells. Cell viability was determined by an MTT assay, and the released NO was measured with a Griess reagent. (**A**) Cell viability. (**B**) Nitrite release. The data represent the mean ± SD. **** *p* < 0.0001 vs. Veh (vehicle, PBS); #### *p* < 0.0001 vs. LPS-treated group (*n* = 3). One-way ANOVA test, followed by Dunnett’s test. “ns” represents no statistical difference.

**Figure 2 ijms-25-10594-f002:**
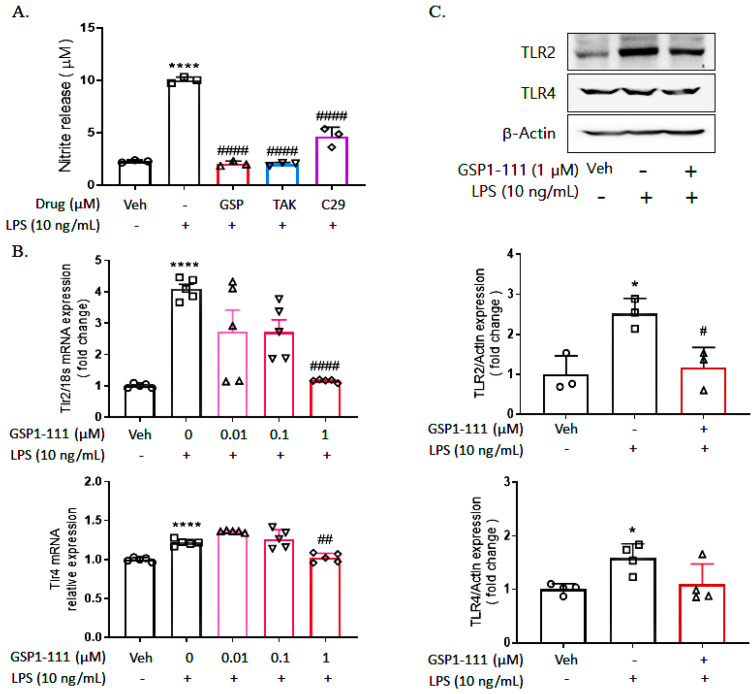
GSP1-111 reduces the mRNA expression of TLR2 in BV2 microglial cells. BV2 microglial cells were treated with GSP1-111 (1 μM), TAK-242 (a specific inhibitor of TLR4, 0.5 μM), and C29 (a specific inhibitor of TLR2, 50 μM) for 1 h. After 24 h, nitrite release was measured using Griess reagent (**A**). BV2 microglial cells were treated with GSP1-111 (0.01, 0.1, and 1 μM), and then 10 ng/mL of LPS was added to the cultured cells after 1 h. After 24 h, the mRNA levels of TLR2 and TLR4 were measured by RT-qPCR (**B**), and the protein expression of TLR2/4 was measured by Western blot analysis (**C**). The data represent the mean ± SD. * *p* < 0.05, **** *p* < 0.0001 vs. Veh (vehicle, PBS); # *p* < 0.05, ## *p* <0.01, #### *p* < 0.0001 vs. LPS-treated group (*n* = 3–5). One-way ANOVA test, followed by Dunnett’s test.

**Figure 3 ijms-25-10594-f003:**
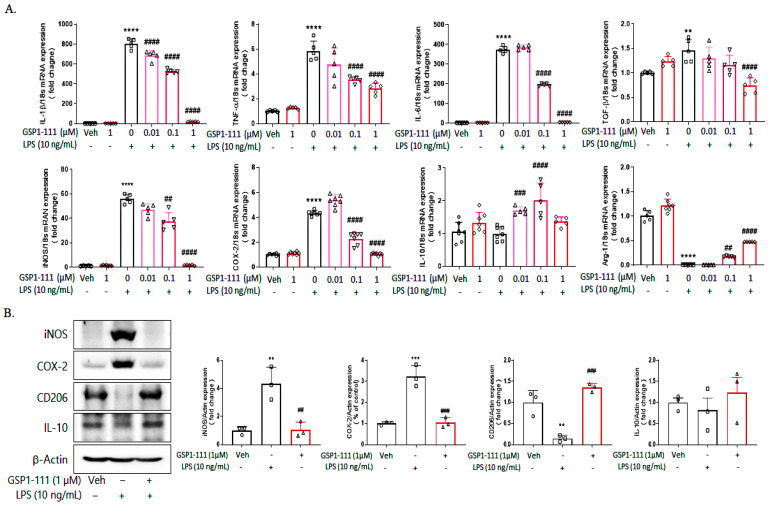
GSP1-111 treatment significantly decreases the M1-specific inflammatory markers in BV2 microglial cells. (**A**) The mRNA expression of M1-specific markers (IL-1β, TNF-α, IL-6, TGFβ, iNOS, and COX-2) and M2-specific markers (IL-10 and Arg-1) (*n* = 5–7), (**B**) The protein expression of M1-specific (iNOS and COX-2) and M2-specific markers (CD206 and IL-10) (*n* = 3). The data represent the mean ± SD. ** *p* < 0.01, *** *p* < 0.001, **** *p* < 0.0001 vs. Veh (vehicle, PBS); ## *p* < 0.01, ### *p* < 0.001, #### *p* < 0.0001 vs. LPS-treated group. One-way ANOVA test followed by Dunnett’s test.

**Figure 4 ijms-25-10594-f004:**
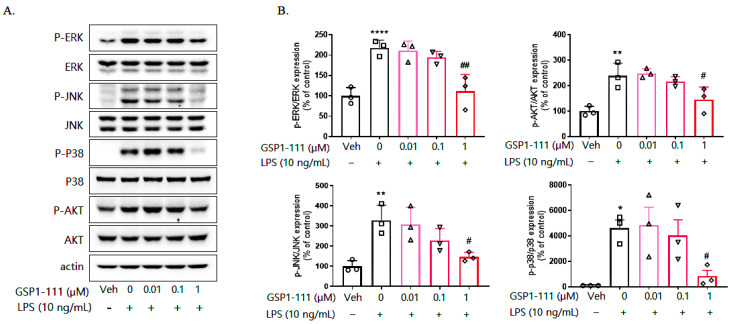
GSP1-111 downregulates LPS-induced MAPKs and Akt phosphorylation. (**A**) BV2 microglial cells were treated with GSP1-111 (0.01, 0.1, and 1 μM) for 1 h, then 10 ng/mL of LPS was added to the cultured cells. After 30 min, cells were harvested to measure the phosphorylation level of MAPK proteins using Western blot analysis. (**B**) The data represent the mean ± SD.* *p* < 0.05, ** *p* < 0.01, **** *p* < 0.0001 vs. Veh (vehicle, PBS); # *p* < 0.05, ## *p* < 0.01 vs. LPS-treated group (*n* = 3). One-way ANOVA test, followed by Dunnett’s test.

**Figure 5 ijms-25-10594-f005:**
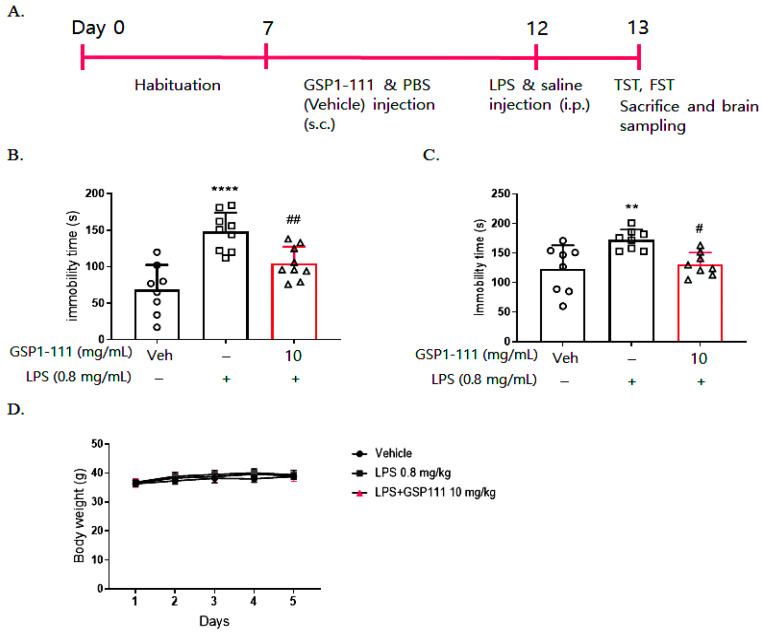
GSP1-111 decreases LPS-induced depression-like behavior. (**A**) The experimental timeline. Mice were injected with GSP1-111 (10 mg/kg, s.c) and PBS (vehicle) for 5 days, and on the last day, mice were injected with LPS (0.8 mg/kg, i.p) and saline (vehicle control). After 24 h, a forced swim test and tail suspension test were performed, and immobility time was measured (*n* = 9–10). (**B**) Immobility time in FST. (**C**) Immobility time in TST. (**D**) Body weight. The data represent the mean ± SD. ** *p* < 0.01, **** *p* < 0.0001 vs. vehicle control (saline); # *p* < 0.05, ## *p* < 0.01 vs. LPS-injected group. One-way ANOVA test, followed by Dunnett’s test.

**Figure 6 ijms-25-10594-f006:**
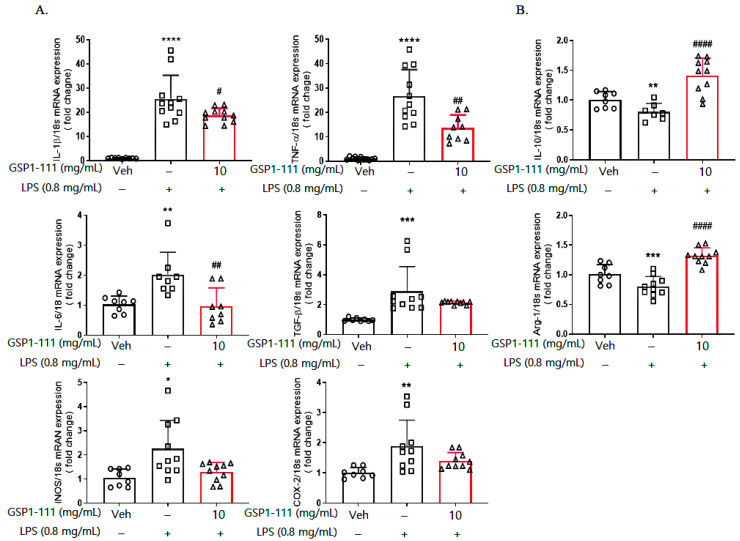
GSP1-111 significantly prevents LPS-induced M1 polarization in the brain. After a forced swim test, the mice were euthanized, and the brain was dissected (*n* = 5). The mRNA expression level of M1-specific markers and M2-specific markers in the frontal cortex of the brain were measured by RT-qPCR. (**A**) M1-specific pro-inflammatory markers. (**B**) M2-specific anti-inflammatory makers. The data represent the mean ± SD. * *p* < 0.05, ** *p* < 0.01, *** *p* < 0.001, and **** *p* < 0.0001 vs. vehicle control (saline); # *p* < 0.05, ## *p* < 0.01, #### *p* < 0.0001 vs. LPS-injected group (*n* = 8–11). One-way ANOVA test followed by Dunnett’s test.

**Figure 7 ijms-25-10594-f007:**
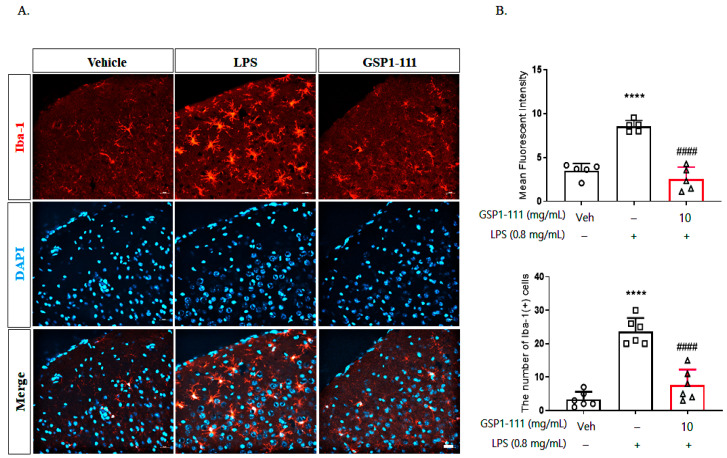
GSP1-111 decreases microglial activation in the brains of LPS-injected mice. After a behavior test, the mice were euthanized, and the brain was perfused for immunohistochemistry analysis (*n* = 5). (**A**) Microglial activation by immunocytochemistry using a microglial-specific antibody, Iba-1(red). Scale bar = 20 μm. (**B**) Iba-1 fluorescence intensity using ImageJ, with the number of Iba-1 (+) cells. The data represent the mean ± SD. **** *p* < 0.0001 vs. vehicle control (saline); #### *p* < 0.0001 vs. LPS-injected group (*n* = 5). One-way ANOVA test, followed by Dunnett’s test.

**Figure 8 ijms-25-10594-f008:**
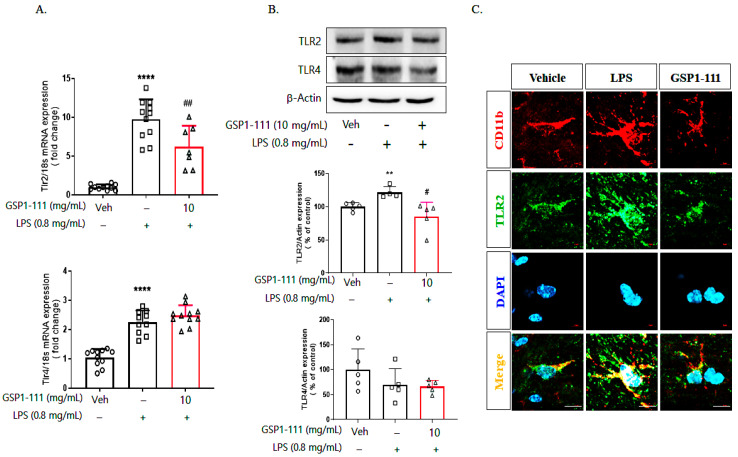
GSP1-111 inhibits the protein expression of TLR2 in the brain. After a forced swimming test, the mice were euthanized, and the brain was dissected. The expression levels of TLR2 and TLR4 mRNA were measured by RT-qPCR. (**A**) mRNA expression level of TLR2 and TLR4 in the frontal cortex of the brain (*n* = 9–11). (**B**) Protein expression level of TLR2 and TLR4 (*n* = 5). (**C**) Immunostaining for TLR2 and CD11b. Scale bar = 10 μm. The data represent the mean ± SD. ** *p* < 0.01, **** *p* < 0.0001 vs. vehicle control (saline); # *p* < 0.05, ## *p* < 0.01, vs. LPS-injected group. One-way ANOVA test, followed by Dunnett’s test.

**Table 1 ijms-25-10594-t001:** Gene-specific primer sequences.

	Sense	Antisense
*IL-1β*	AAA ATG CCT CGT GCT GTC TG	CTA TGT CCC GAC CAT TGC TG
*Il-6*	TTG TGC AATGGC AAT TCT GA	TGG AAG TTG GGG TAG GAA GG
*TNFα*	TAG CCC ACG TCG TAG CAA AC	GGA GGC TGA CTT TCT CCT GG
*TGF* *β*	AGC TGC GCT TGC AGA GAT TA	CAC TTC CAA CCC AGG TCC TT
*iNOS*	CTG GCT GCC TTG TTC AGC TA	AGT GTA GCG TTT CGG GAT CT
*COX-2*	TGCTGTACAAGCAGTGGCAA	AGGTGCTCGGCTTCCAGTAT
*IL-10*	AGGCGCTGTCATCGATTTCT	ATGGCCTTGTAGACACCTTGG
*Arg-1*	ACAAGACAGGGCTCCTTTCAG	CGTTGAGTTCCGAAGCAAGC
*Tlr2*	TGCTTTCCTGCTGGAGATTT	TGTAACGCAACAGCTTCAGG
*Tlr4*	ACCTGGCTGGTTTACACGTC	CTGCCAGAGACATTGCAGAA
*18S rRNA*	CATTAAATCAGTTATGGTTCCTTTGG	TCGGCATGTATTAGCTCTAGAATTACC

## Data Availability

Data are contained within the article.

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
