# Peer review of "GSP1-111 Modulates the Microglial M1/M2 Phenotype by Inhibition of Toll-like Receptor 2: A Potential Therapeutic Strategy for Depression"

_ijms, 2024, doi:10.3390/ijms251910594_

Round 1

Reviewer 1 Report

Comments and Suggestions for Authors

The paper is very interesting because it indicates new potential therapeutic targets not only for neurodegenerative diseases but also for neuropsychiatric diseases.

The authors focused on studying the role of neuroinflammation, especially Toll receptors (TLRs), the regulation of which in microglia may be a potential therapeutic strategy. They tested GSP1-111, a new synthetic peptide inhibiting TLR signaling, on neuroinflammation and depression-like behaviors. The methods used are adequate to the studied subject. While both the introduction and the description of results, as well as the discussion, raise no concerns, I do have several important comments regarding the methods, which I outline below:

I do not understand the rationale behind cropping blots from the original images that have been placed in the supplement. The principle of presenting results transparently requires showing raw data to confirm that they have not been manipulated. Please provide full images of the blots.

Please mark individual subjects on all graphs, as has been done, for example, in Figure 5.

The description of the control used in the experiments is unclear. In what solvent is the GSP1-111 compound dissolved? Is the control saline or PBS buffer? Similarly, the method of administration varies. How, then, is the control treated? This is critical for correctly interpreting the results.

For the Discussion section:

It is recommended to perform experiments on both male and female subjects. Please include a brief section discussing potential changes in females. Do the authors expect sex-related differences?

Comments on the Quality of English Language

Moderate editing of the English language is required.

Author Response

Reviewer 1;

First of all, we would like to thank you for handling our manuscript. We appreciate the important points raised by this comment. We believe this comment helped a lot to improve the quality of the manuscript. We added this point throughout the manuscript including introduction, method, result and discussion. We tried to answer the suggestions raised by reviewers as faithfully as possible. We believe that expert comments from the reviewers helped a lot to improve the quality of the present works.

Point 1. I do not understand the rationale behind cropping blots from the original images that have been placed in the supplement. The principle of presenting results transparently requires showing raw data to confirm that they have not been manipulated. Please provide full images of the blots.

Response 1:

It seems that the protein marker figure on the left side, included to explain the protein markers we used, has caused some confusion. After performing SDS-PAGE, we provided the blot image where the original blot was cut according to the size of the target protein, followed by binding with primary antibodies and detection. This is a more detailed figure based on the blot we provided.

We used full blots for the Western blot experiments. After verifying the molecular weight markers, we accurately cut the blots according to the molecular weight of the target proteins and incubated them with the primary antibodies specific to the protein. This method is commonly used by many researchers because it allows for accurate detection of target protein at the precise molecular weight. Additionally, this approach allows for the simultaneous detection of control proteins like b-actin, which is cut from the same membranes with different molecular weight, making comparative analysis more accurate than using separate blots, which may cause the variation by loading errors etc.

The blots we provided are the original blots containing all the groups used in this experiment’s analysis. They were obtained using the LAS3000 image detection system and are raw data with no manipulation like cropping and reattachment. From a technical standpoint, we believe this method is more accurate for confirming the expression of the target proteins without unnecessary loss of the sample and reagents.

Although we regret not being able to provide the entire blot, we assure you that the blots used in this experiment are the original blots without any artificial modification.

Point 2. Please mark individual subjects on all graphs, as has been done, for example, in Figure 5.

Response 2: As per your kind comments, we marked individual subjects on all graphs.

Point 3. The description of the control used in the experiments is unclear. In what solvent is the GSP1-111 compound dissolved? Is the control saline or PBS buffer? Similarly, the method of administration varies. How, then, is the control treated? This is critical for correctly interpreting the results.

Response 3: Thank you for your helpful comments. We used PBS as the vehicle for the subcutaneous administration of GSP1-111, and saline was used as the control for the intraperitoneal administration of LPS.

As per your comments, we have added the vehicle and the method of administration in Section 2.5 of the results as follows, and we described the vehicle as either PBS or saline in all figure legends. 

  • Line 223-224

“After administering GSP1-111 and PBS (vehicle) subcutaneously for 5 days, LPS and saline (control) were given intraperitoneally.”

We added this point in figure 5A as follows.

Point 4. For the Discussion section:

 It is recommended to perform experiments on both male and female subjects. Please include a brief section discussing potential changes in females. Do the authors expect sex-related differences?

Response 4: Thank you for your helpful comment. We used male mice in this study as an initial step to investigate the effect of GSP1-111.We are very well aware of the importance of biological sex in biomedical studies and have long been interested in sex-related differences in neurological disorders, including depression, Alzheimer’s disease, neurodevelopmental disorders such as autism spectrum disorders and ADHD, which have been published in several articles [1-5]. At the moment, there is no theoretical and experimental clues that there might be sex difference in the effects of GSP1-111 between male and female, but we will carry out the same study in female and compare the results with those obtained from male animals in the future. We greatly appreciate the helpful comment.

As per your comment, we added this point in discussion section as follows:

  • Discussion

“Depression is a common condition that significantly impairs psychosocial functioning and lowers quality of life [6, 7]. Women are more vulnerable than men to stress-related psychopathologies such as depression, with twice the incidence rate and four times the risk of experiencing recurrent depressive episodes. Among the various symptoms associated with mood disorders in women, depression is reported to be the most severe. Chronic stress is recognized as a major risk factor for major depressive disorder (MDD), affecting the immune system and activating microglia in the mPFC. Our experimental model is an acute depression model based on inflammation. Initially, we tested the GSP1-111 compound on male animals to evaluate its efficacy against depression. It is plausible that the differences between males and females may not be evident in the inflammatory-induced depression animal model we used. Reports suggest behavioral differences and varying drug responses between males and females in long-term depression models and human depression studies[8-10]. Recent studies have shown that both male and female mice exposed to chronic stress exhibit depression-like behaviors, but only female mice display persistent depression-like behaviors [11]. Moreover, this persistent depressive behavior in female mice has been linked to TLR4 and microglial activation. This study supports the hypothesis that TLR4 in microglia may regulate sex differences in persistent depression-like behaviors in females. Additionally, TLR4 knockout mice exhibited pronounced depressive-like behavior, while TLR2 knockout mice showed significant impairment in recovery from depression in male mice[1]. There is also evidence that anxiety and social avoidance are induced by microglial activation through TLR2/4 in a repeated social defeat stress-induced depression model[12]. Therefore, future studies are needed to investigate inflammation and sex-related differences in neurological disorders such as depression.”

Thank you in advance for your consideration of this manuscript. I look forward to hearing from you.

Sincerely yours,

Kyoung Ja Kwon, PhD

Reviewer 2 Report

Comments and Suggestions for Authors

This is an interesting paper showing that a novel peptide (GSP1-111) has the potential to mitigate depression-like behavior in mice by inhibiting TLR2 and reducing inflammatory responses. In general, the Introduction, Methods, Results, and Discussion sections are relevant and presented in an appropriate manner. I would like to highlight a few issues that could be addressed to enhance the quality of the manuscript.

- Introduction (Lines 76-79): Here the authors present some information about the structural characteristics of TLRs. Is there data on the interactions between GSP1-111 and the TLRs, whether in silico or otherwise, that would be relevant to comment in this paper? Besides, please also provide reference(s) to support the statement in Lines 108-109.

- Introduction (Lines 85-86): Please rephrase these lines as it appears that the authors mentioned lipopolysaccharide (LPS), found for example in gram-negative bacteria, as a DAMP rather than a PAMP.

- Is there information available regarding the permeability of GSP1-111 through the BBB? If so, please add in the Discussion if appropriate.

- Figure legends: There are some incorrect symbols in the figure legends and their use should be revised throughout the document.

- Figure 5: The n reported in the figure legend is n=10, which is not consistent with the dots represented in the figure. Also, in section 4.2. (Animals; Line 414) the authors report an n of 9 animals per group. Please carefully revise this issue. In addition, please report the n in all figure legends.

- Figure 7: Please include the merged figures of IBA-1 and DAPI staining.

- Figures: Please increase the quality of figures for publication.

- Section 4.2 (Animals): Please include the total number of animals used in the study.

- Provide references for the doses and concentrations of LPS and GSP1-111 used in the protocols.  

- Statistical analysis: Please include the post hoc test.

- Funding (Line 569): Please provide the funding information, if any.

- Please replace the term "sacrificed" with "euthanized" in the text and figures.

Author Response

Reviewer 2;

First of all, we would like to thank you for handling our manuscript. We appreciate the important points raised by this comment. We believe this comment helped a lot to improve the quality of the manuscript. We added this point throughout the manuscript including introduction, method, result and discussion. We tried to answer the suggestions raised by reviewers as faithfully as possible. We believe that expert comments from the reviewers helped a lot to improve the quality of the present works.

Point 1. Introduction (Lines 76-79): Here the authors present some information about the structural characteristics of TLRs. Is there data on the interactions between GSP1-111 and the TLRs, whether in silico or otherwise, that would be relevant to comment in this paper? Besides, please also provide reference(s) to support the statement in Lines 108-109.

Response 1:

We provided a brief information regarding GSP1-111 in the introduction section.

GSP1-111 is being developed as a new drug for autoimmune disease by Genesen Co., Ltd. GSP1-111 targets TLRs, specifically TLR4, as a therapeutic agent for autoimmune diseases such as rheumatoid arthritis and lupus. Although the binding result could not be included in the results, it has been confirmed using surface plasmon resonance (SPR) that GSP1 (light brown) binds to the BB loop (red) of the TIR domain of TLR4-1198 [1]. This binding interaction suggests a potential mechanism by which GSP1 can modulate TLR4 signaling pathways.

<The surface plasmon resonance (SPR) result. This figure shows that GSP-1 (light brown) binds to the BB loop (710-RDFIPG- VAIAA-720, red) of the TIR domain of TLR4-1198. >

According to your comments, we added this point and the references to our material property in Line 107-111 of introduction. Thank you for your helpful comments.

  • Line 107

GSP1-111 is one of the peptide series designed by binding cell-penetrating peptide (CPP) to eight sequences within the TIR domain of TIRAP, an adaptor protein of TLR. This peptide, consisting of 26 amino acids, penetrates into the cells and blocks the interaction between the intracellular Toll-IL-1 receptor (TIR) domain of TLR and the adaptor protein TIRAP, thereby inhibiting TLR-mediated downstream signaling and effectively disrupting TLR4 signal transduction [41-43].

  1. Thapa, B., et al., Cell-penetrating TLR inhibitor peptide alleviates ulcerative colitis by the functional modulation of macrophages. Front Immunol, 2023. 14: p. 1165667.
  2. Kwon, H.K., et al., A cell-penetrating peptide blocks Toll-like receptor-mediated downstream signaling and ameliorates autoimmune and inflammatory diseases in mice. Exp Mol Med, 2019. 51(4): p. 1-19.
  3. Jung, S.W., et al., A Cell-Penetrating Peptide That Blocks Toll-Like Receptor Signaling Protects Kidneys against Ischemia-Reperfusion Injury. Int J Mol Sci, 2021. 22(4).

Point 2. Introduction (Lines 85-86): Please rephrase these lines as it appears that the authors mentioned lipopolysaccharide (LPS), found for example in gram-negative bacteria, as a DAMP rather than a PAMP.

Response 2: Thank you for your careful comment. We rephrase these sentences as follow.

  • Line 84-90

“TLRs can be activated by different DAMPs, including endogenous molecules released from damaged cells such as HMGB1, heat shock proteins, nucleic acid (dsRNA, dsDNA, ssRNA), and mitochondrial DNA (mtDNA). In addition, TLRs are activated by the binding of PAMPs, which include lipopolysaccharide (LPS) from gram-negative bacteria, flagellin, bacterial/viral nucleic acid including double-stranded RNA (dsRNA), and single-stranded RNA (ssRNA), CpG rich DNA, b-glucan from fungus, and trehalose dimycolate (TDM) from Mycobacterium [31,32].”

Point 3. Is there information available regarding the permeability of GSP1-111 through the BBB? If so, please add in the Discussion if appropriate.

Response 3: We appreciate your valuable comment. GSP1-111 was supplied by Genesen Co., Ltd. (Seoul, Korea) as a novel synthetic peptide. GSP1-111 is one of peptide series consisting of 26 amino acids, designed to penetrate cells by being tagged with a cell-penetrating peptide (CPP). CPPs are a specialized group of peptides capable of crossing cell membrane bilayers without causing a significant lethal membrane damage. Blood-brain barrier (BBB) penetration is a pivotal challenge in the development of treatments for neurological disorders, particularly nucleic acid-based medicines. CPPs stand out as the most promising technology for enabling macromolecules, such as peptides, to penetrate the BBB. Some studies have shown that CPPs can reach the brain parenchyma both in vivo and in vitro [2, 3]. While we do not yet have direct evidence that GSP1-111 can cross the BBB, its clear effect on neuroinflammation suggests efficient permeation of GSP-111 through the BBB. Currently Genesen Co. Ltd., is conducting a PK study for the clinical development of GSP1-111, and expects to obtain pharmacokinetic data regarding the penetration of GSP-111 into the brain. 

As per your valuable comments, we added this point in discussion section as follows.

  • Line 340-351

GSP1-111 was supplied by Genesen Co., Ltd. (Seoul, Korea) as a novel synthetic peptide. GSP1-111 is one of peptide series consisting of 26 amino acids, designed to penetrate cells by being tagged with a cell-penetrating peptide (CPP). CPPs are a specialized group of peptides capable of crossing cell membrane bilayers without causing a significant lethal membrane damage. Blood-brain barrier (BBB) penetration is a pivotal challenge in the development of treatments for neurological disorders, particularly nucleic acid-based medicines. CPPs stand out as the most promising technology for enabling macromolecules, such as peptides, to penetrate the BBB. Some studies have shown that CPPs can reach the brain parenchyma both in vivo and in vitro [2, 3]. While we do not yet have direct evidence that GSP1-111 can cross the BBB, its clear effect on neuroinflammation suggests efficient permeation of GSP-111 through the BBB. Further studies on BBB penetration, including brain pharmacokinetics, are needed.

References

  1. Kwon, H.K., et al., A cell-penetrating peptide blocks Toll-like receptor-mediated downstream signaling and ameliorates autoimmune and inflammatory diseases in mice. Exp Mol Med, 2019. 51(4): p. 1-19.
  2. Adil, K.J., et al., Behavioral Deficits in Adolescent Mice after Sub-Chronic Administration of NMDA during Early Stage of Postnatal Development. Biomol Ther (Seoul), 2022. 30(4): p. 320-327.
  3. Zou, L.L., et al., Cell-penetrating Peptide-mediated therapeutic molecule delivery into the central nervous system. Curr Neuropharmacol, 2013. 11(2): p. 197-208.

Point 4. Figure legends: There are some incorrect symbols in the figure legends and their use should be revised throughout the document.

Response 4: we appreciate your helpful comment. We corrected this point throughout this manuscript. 

Point 5. Figure 5: The n reported in the figure legend is n=10, which is not consistent with the dots represented in the figure. Also, in section 4.2. (Animals; Line 414) the authors report an n of 9 animals per group. Please carefully revise this issue. In addition, please report the n in all figure legends.

Response 5: Thank you for your helpful comment. 9 animals per group is based on the sample size calculation for animal experiment. Based on the sample size calculation, 9 and 10 animals were used in each behavioral test. The graph presents the data after removing outliers using the 1.5IQR method.

As per your careful comment, we added n in all figure legends.  

Point 6. Figure 7: Please include the merged figures of IBA-1 and DAPI staining.

Response 6: As per your comment, we added the merged figures of IBA-1 and DAPI staining in Figure 7A as follows.

Point 7. Figures: Please increase the quality of figures for publication.

Response 7: As per your comment, we tied to improve the resolution ratio of figures and changed these figures.

Point 8. Section 4.2 (Animals): Please include the total number of animals used in the study.

Response 8: As per your comments, we added the total number of animals used in this study of section 4.2. as follows:

  • Line 465-467

“A total of 19 male ICR mice (20–30 g; four weeks old) were purchased from Orient Bio (Gyeonggi, Korea) and used for the behavior experiments.”

Point 9. Provide references for the doses and concentrations of LPS and GSP1-111 used in the protocols.  

Response 9: Thank you for your careful comments.

As a result of determining the IC50 for TNFa, IL-1b and IL-6 in mouse Raw264.7 cells, the IC50 values for TNFa and IL-1b were 0.13 mM, and for IL-6, it was 0.08 mM. Therefore, we selected concentrations of 0.01, 0.1 and 1 mM for this experiment based on the IC50 values. In addition, these concentrations (0.01, 0.1, and 1 mM) did not show any cytotoxicity in BV-2 microglial cells. As our results demonstrated, 1 mM of GSP1-111 reduced the expression of inflammatory mediators, such as NO, almost to the control level, so higher concentrations were not tested. According to Genesen Co., Ltd., the developer of this substance, no cytotoxicity was observed even at a concentration of 10 mM of this substance.

The dose of LPS used in this study was 0.8 mg/kg according to our previous paper and other paper. We added these references in section 4.8. as follow.

  • Line 552

“On day 5, mice were injected with 0.8 mg/kg of LPS or saline intraperitoneally at a volume of 10 mL/kg [74,75].”

Point 10. Statistical analysis: Please include the post hoc test.

Response 10: We added the post hoc test in section 4.12. statistical analysis and rephrased these sentences as follow. Thank you for your helpful comment.

“Data were analyzed using GraphPad Prism Version 8.4.3 software (GraphPad Software, Inc., San Diego, CA, USA). All data were expressed as the mean ± standard deviation (SD). Statistical comparisons were performed using one-way analysis of variance (ANOVA) test followed by Dunnett’s multiple comparison test as a post hoc test and a value of P< 0.05 was considered significant.

Point 11. Funding (Line 569): Please provide the funding information, if any.

Response 11: We have updated both the acknowledgement statement and the funding information, line 617 and 627. Thank you for kind comment.

Point 12. Please replace the term "sacrificed" with "euthanized" in the text and figures.

Response 12: We replace the term “sacrificed” to “euthanized” in the text and figures.

Thank you in advance for your consideration of this manuscript. I look forward to hearing from you.

Sincerely yours,

Kyoung Ja Kwon, PhD

Round 2

Reviewer 1 Report

Comments and Suggestions for Authors

Thank you for clarifying and improving the manuscript.